# Serotypes, Antimicrobial Resistance Profiles, and Virulence Factors of *Salmonella* Isolates in Chinese Edible Frogs (*Hoplobatrachus rugulosus*) Collected from Wet Markets in Hong Kong

**DOI:** 10.3390/foods12112245

**Published:** 2023-06-01

**Authors:** Sara Boss, Roger Stephan, Jule Anna Horlbog, Ioannis Magouras, Violaine Albane Colon, Kittitat Lugsomya, Marc J. A. Stevens, Magdalena Nüesch-Inderbinen

**Affiliations:** 1Institute for Food Safety and Hygiene, Vetsuisse Faculty, University of Zurich, 8057 Zurich, Switzerland; sara.boss@uzh.ch (S.B.); stephanr@fsafety.uzh.ch (R.S.); jhorlbog@fsafety.uzh.ch (J.A.H.); marc.stevens@uzh.ch (M.J.A.S.); 2Department of Infectious Diseases and Public Health, Jockey Club College of Veterinary Medicine and Life Sciences, City University of Hong Kong, Kowloon, Hong Kong SAR, China; ioannis.magouras@cityu.edu.hk (I.M.);; 3Centre for Applied One Health Research and Policy Advice, Jockey Club College of Veterinary Medicine and Life Sciences, City University of Hong Kong, Kowloon, Hong Kong SAR, China

**Keywords:** wet markets, edible frogs, *Salmonella*, antimicrobial resistance, ampicillin, azithromycin, ciprofloxacin, *bla*
_DHA-1_, *mphA*, *qnr*

## Abstract

*Salmonella* is an important agent of gastrointestinal disease in humans. While livestock, such as cattle, poultry, and pigs, are well-recognised animal reservoirs of *Salmonella*, there is a lack of data on *Salmonella* in edible frogs, even though frog meat is a popular food worldwide. In this study, 103 live edible Chinese frogs (*Hoplobatrachus rugulosus*) were collected from wet markets throughout Hong Kong. After euthanasia, faeces or cloacal swabs were examined for *Salmonella*. Overall, *Salmonella* spp. were isolated from 67 (65%, CI: 0.554–0.736) of the samples. The serotypes included *S*. Saintpaul (33%), *S*. Newport (24%), *S*. Bareilly (7%), *S*. Braenderup (4%), *S*. Hvittingfoss (4%), *S*. Stanley (10%), and *S*. Wandsworth (16%). Many isolates were phylogenetically related. A high number of genes encoding for resistance to clinically relevant antimicrobials, and a high number of virulence determinants, were identified. Antimicrobial susceptibility testing (AST) identified multidrug resistance (MDR) in 21% of the isolates. Resistance to ampicillin, ciprofloxacin, nalidixic acid, and tetracycline was common. These results demonstrate that a high percentage of live frogs sold for human consumption in wet markets are carriers of multidrug-resistant *Salmonella*. Public health recommendations for handling edible frogs should be considered, to mitigate the risk of *Salmonella* transmission to humans.

## 1. Introduction

Non-typhoidal *Salmonella* (NTS) species are among the most important etiological agents of gastrointestinal diseases in humans worldwide, causing an estimated 180 million diarrhoeal illnesses each year [1]. Most human infections involve the *Salmonella enterica* subspecies *enterica,* which contains more than 2600 serotypes (serovars) [2]. *Salmonella* serotypes are assigned using the traditional antibody-based serological method, according to the Kauffmann–White–Le Minor scheme [3], or by applying DNA-sequenced-based typing methods, such as multilocus sequence typing (MLST) and genetic antigen prediction using whole genome sequencing (WGS) data [4,5].

The five most prevalent *Salmonella* serotypes involved in culture-confirmed human salmonellosis cases in the European Union in 2021 were *S.* Enteritidis, *S.* Typhimurium, monophasic *S.* Typhimurium 4,[5],12:i:−, *S.* Infantis, and *S.* Derby [6]. The five most prevalent *Salmonella* serotypes involved in culture-confirmed human salmonellosis cases in the United States in 2016 were *S.* Enteritidis, *S.* Newport, *S.* Typhimurium, *S.* Javiana, and monophasic *S.* Typhimurium 4,[5],12:i:− [7,8]. The five most prevalent *Salmonella* serotypes involved in culture-confirmed human salmonellosis cases in China, between 2014 and 2021 were *S.* Typhimurium, *S.* Enteritidis, monophasic *S*. Typhimurium 4,[5],12:i:-, *S.* London, and *S.* Stanley [9].

Typically, salmonelloses present as self-limiting episodes; however, severe cases of infection—including bacteraemia and meningitis—require antimicrobial treatment [10]. Ciprofloxacin is a common first-line antimicrobial for treating salmonellosis, but because fluoroquinolones are not used for treating children, β-lactams (ampicillin or third-generation cephalosporins) and azithromycin are of equal importance [11]. Antimicrobial resistance (AMR) in *Salmonella* is associated with higher morbidity and mortality, compared to susceptible strains, and is therefore a major concern for public health [12]. As a consequence of the global increase of AMR in NTS, fluoroquinolone-resistant *Salmonella* feature on the WHO high priority list of resistant pathogens for which research and development of new antimicrobials are urgently required [13].

The most common reservoir for *Salmonella* spp. is the intestinal tract of domestic and wild animals, including cattle, pigs, and poultry [7]: accordingly, a variety of food matrices, such as beef, pork, poultry meat, raw eggs, and seafood may serve as vehicles for the transmission of susceptible and resistant *Salmonella* to humans [14].

Frog meat is an important source of protein in many African and Asian countries [15]. The Chinese edible bullfrog (*Hoplobatrachus rugulosus*) is a large, amphibian species, mainly found in Asian countries, such as Cambodia, China, the Philippines, Taiwan, Thailand, and Vietnam [15]. *H. rugulosus* is widely farmed or harvested in the wild in several Asian countries, and is sold for human consumption in the markets of these countries, or is traded internationally as frogs’ legs, in large numbers [16]. Between 2010 and 2019, the EU imported an estimated 40,700 tonnes of frogs’ legs, corresponding to about 814–2000 million individual frogs, with a large amount of these animals imported from Asia [15,17]. 

Despite the popularity of frogs among consumers worldwide, and the economic importance to many countries of the global trade in frogs’ legs, data on edible frogs as possible reservoirs for foodborne pathogens such as *Salmonella* are currently lacking.

Therefore, this pilot study aimed to assess the occurrence of *Salmonella* in *H. rugulosus* sold for human consumption in wet markets in Hong Kong, and to characterise the isolates, using phenotypic and genotypic methods, including serotyping, antimicrobial susceptibility testing, virulence profiling, and whole genome analysis.

## 2. Materials and Methods

### 2.1. Sample Collection

All officially registered wet markets in Hong Kong (a total of 94 markets distributed across Hong Kong, including Hong Kong Island, Kowloon, and the New Territories) [18] were visited twice by a research assistant within a period of three months, and markets that sold edible frogs were noted. Of these wet markets, a total of nine (designated A–I) were selected randomly, and a total of 103 live Chinese edible frogs (*H. rugulosus*) were collected during March and April 2022 (Table 1). The country of origin of the frogs from each market was recorded from the vendors.

The collected frogs were taken to the Department of Infectious Diseases and Public Health, City University of Hong Kong. Physical examinations of the frogs were performed by a certified veterinarian, and age, sex, and weight were documented. For this study, all frogs were considered, regardless of their age and sex. 

Anaesthesia was induced by MS-222 (Tricaine methanesulfonate, Syndel, Ferndale, WA, USA) bathing at a concentration of 2 g/L. The MS-222 was buffered by an equal concentration of sodium bicarbonate (Sigma-Aldrich, St. Louis, MO, USA). Once anaesthesia was confirmed, humane euthanasia was induced by an intravenous injection of pentobarbital (Dorminal 20%, Alfasan, Woerden, The Netherlands) at a dose of 100 mg/kg in the ventral abdominal vein. Decapitation was performed as a secondary method of euthanasia, following the guidelines of the American Veterinary Medical Association [19]. During post-mortem examination, faeces or cloacal swabs were aseptically collected, and placed into sterile tubes containing Amies agar gel (Thermo Fisher Scientific, Melbourne, Australia), until further processing. 

### 2.2. Salmonella Isolation and Identification 

*Salmonella* spp. detection was performed, using the ISO 6579-1: 2017 method for isolation and identification of *Salmonella* [20]. Each swab was placed in 10 mL buffered peptone water (BPW; Thermo Fisher Scientific), and incubated at 37 °C for 18 h. Following incubation, 0.1 mL of cultured BPW was used to inoculate 10 mL Rappaport-Vassiliadis Soya Peptone broth (RVS broth; bioMérieux, Marcy-l’Étoile, France), with incubation at 41.5°C for 24 h. Cultured RVS broth was streaked on Xylose Lysine Deoxycholate agar (XLD agar; Thermo Fisher Scientific), and incubated at 37 °C for 24 h. *Salmonella* Typhimurium ATCC 14028 was used as a positive control. No plating media, other than XLD, were used, and no other biochemical characterisations were done. Putative *Salmonella* colonies with black morphologies were subjected to species identification by Matrix-assisted laser desorption/ionisation time-of-flight mass spectrometry (MALDITOF; Bruker, MA, US), and analysed using the MALDI Biotyper (Bruker, MA, US). For calibration and internal quality control of the MALDI Biotyper System, the Bruker Bacterial Test Standard (BTS) containing extract of *Escherichia coli* DH5 alpha was used, according to the manufacturer’s instructions (Bruker). *Salmonella* Typhimurium ATCC 14028 was used for internal evaluation.

### 2.3. Salmonella Serotyping 

All isolates were serotyped at the Swiss National Reference Centre for Enteropathogenic Bacteria and *Listeria* (NENT), Switzerland. Serotyping was performed according to the Kauffmann–White–Le Minor scheme [3], using commercially available antisera (Sifin Diagnostics, Berlin, Germany).

### 2.4. DNA Extraction and Whole Genome Sequencing 

Isolates were grown on sheep blood agar at 37 °C overnight, prior to DNA isolation using the DNeasy Blood & Tissue Kit (Qiagen, Hombrechtikon, Switzerland). DNA libraries were prepared, using a Nextera DNA Flex Sample Preparation Kit (Illumina, San Diego, CA, USA). Whole genome sequencing was performed on an Illumina MiniSeq Sequencer (Illumina). The Illumina reads files passed the standard quality checks, using the software package FastQC 0.11.7 (Babraham Bioinformatics, Cambridge, UK), and were assembled using the Spades 3.14.1-based software Shovill 1.0.4 [21], using default settings. The assembly was filtered, retaining contigs >500 bp, and annotated using the NCBI prokaryotic genome annotation pipeline [22]. 

Whole-genome-based *Salmonella* serotyping was performed, using SeqSero with standard settings [23]. Sequence types (STs) were determined using a 7-house-keeping gene-based in silico MLST scheme and the Ridom SeqSphereC + software version 8.5 (Ridom GmbH, Münster, Germany). The genetic relatedness of the isolates was assessed by core genome MLST (cgMLST) analyses based on a 3002-locus cgMLST scheme using Ridom SeqSphere+. A minimum spanning tree (MST) was generated for visualisation, with the threshold for cluster identification set to ≤10 alleles between a pair of neighbouring isolates, according to the Ridom SeqSphere+ software. Antimicrobial resistance genes were determined by the Resistance Gene Identifier (RGI) 4.2.2, using database version 3.2.6 [24]. Virulence factors (VFs) were identified by a bi-directional best-hit approach, using Diamond with standard setting [25]⁠. The Prokka-predicted proteomes were compared to the representative-proteins data set A of the virulence factor database VFDB, downloaded in May 2023 [26]⁠. The aggregate VF score was defined as the number of unique VF detected for each isolate.

### 2.5. Antimicrobial Susceptibility Testing 

Antimicrobial susceptibility tests were performed, using the disk-diffusion method, according to the guidelines of the CLSI [27]. Antimicrobial agents included beta-lactams including penicillins and cephalosporins (ampicillin (AM), amoxicillin/clavulanic acid (AMC), cefazolin (CZ), cefotaxime (CTX), cefepime (FEP)), quinolones including fluoroquinolones (nalidixic acid (NA), ciprofloxacin (CIP)), aminoglycosides (gentamicin (G), kanamycin (K), streptomycin (S)), a sulfonamide (sulfamethoxazole–trimethoprim (SXT)), a phosphonic antibiotic (fosfomycin (FOS)), a macrolide (azithromycin (AZM)), a nitrofuran (nitrofurantoin (F/M)), a tetracycline antibiotic (tetracycline (TE)), and an amphenicol (chloramphenicol (C)) (Becton, Dickinson, Heidelberg, Germany). The MICs of the carbapenems ertapenem (ETP), imipenem (IP), and meropenem (MP) for the *bla*_IND-6_-carrying *S*. Wandsworth ST1498 (isolate F33) were determined using ETest^®^ strips (bioMérieux, Marcy-l’Étoile, France). The results were interpreted according to CLSI breakpoints for human clinical isolates, and the isolates were classified as susceptible (S), intermediate (I), or resistant (R) [27]. In the absence of clinical breakpoints for AZM resistance for *Salmonella* other than *S*. *enterica* serotype Typhi, a zone diameter of ≤ 12 mm was interpreted as resistant [27]. Multidrug resistance (MDR) was defined as resistance to at least three antimicrobials from different classes [28].

### 2.6. Statistical Analysis

Comparisons of the proportions of samples containing *Salmonella* from frogs of different age categories (adult or juvenile) and of different sex (male or female) were performed by Fisher’s exact test. The significance criterion was set at *p* ≤ 0.05. Calculations were performed using GraphPad (https://www.graphpad.com, accessed on 8 March 2023).

### 2.7. Ethics Statement

This study was approved of by the Animal Research Ethics Sub-Committee of the City University of Hong Kong (Internal Ref: A- 0698).

## 3. Results

### 3.1. Demographic Data of H. rugulosus Collected from Different Wet Markets

Of the 103 collected frogs, 76 (74%) were adult, and 27 (26%) were juvenile animals. A total of 81 (79%) were male, 21 (20%) were female, and 1 (1%) was a hermaphrodite. The median weight was 164 g (range 91–292 g). Frogs purchased in wet markets A–H were imported from mainland China, whereas those from market I were imported from Thailand. 

### 3.2. Prevalence of Salmonella among H. rugulosus 

*Salmonella enterica* subsp. *Enterica* was isolated from 67 (65%, 95% CI: 0.554–0.736) of the frogs (Table 1). Frogs harbouring *Salmonella* were collected from all nine wet markets, and the proportions of *Salmonella*-positive animals varied, from 17% of the animals collected from wet market C, to 100% of those from markets D, E, and H (Table 1). 

The proportions of adult and of juvenile frogs that tested positive for *Salmonella* were 48/76 (63%) and 19/27 (70%), respectively—a difference that was not statistically significant (*p* = 0.8600). The proportions of male and female frogs harbouring *Salmonella* were 50/81 (62%) and 16/21 (76%), respectively: again, the difference was not significant (*p* = 0.5742).

### 3.3. Serotypes

A total of seven different *Salmonella enterica* subsp. *Enterica* serotypes were identified, using the Kauffmann–White–Le Minor scheme, and were later confirmed by WGS and serotyping by the SeqSero scheme (Table 1). Among the 67 isolates, *S*. Saintpaul (*n =* 22) and *S*. Newport (*n =* 16) were the predominant serotypes, accounting for 33% and 24% of all *Salmonella* isolates, respectively. Other serotypes included *S*. Bareilly (*n =* 5; 7%), *S*. Braenderup (*n =* 3; 4%), *S*. Hvittingfoss (*n =* 3; 4%), *S*. Stanley (*n* = 7; 10%), and *S*. Wandsworth (*n =* 11; 16%) (Table 1). 

### 3.4. Core Genome Multilocus Sequence Types (cgMLST) and Phylogenetic Relatedness

Based on WGS data, the 67 isolates were assigned to nine different STs, whereby ST50 (*n =* 22; 33%) and ST1498 (*n =* 11;16%) were predominant (Table 2). Isolates from the same serotype were assigned to the same ST, except for the *S*. Newport isolates, which showed three different MLST patterns corresponding to ST31 (*n =* 8), ST45 (*n =* 7), and ST46 (*n =* 1), respectively (Table 2). The genetic relatedness of the isolates was visualised by constructing a cgMLST-based phylogenetic tree. The isolates grouped according to serotypes and STs, which were separated by clear allelic distances (Figure 1). Isolates belonging to *S*. Bareilly, *S*. Braenderup, and *S*. Hvittingfoss clustered tightly within their respective serotypes and STs. By contrast, the *S*. Newport ST31 isolates formed two different subclusters (subcluster 1 and subcluster 2 in Figure 1), which consisted of five and three isolates, respectively, while the *S*. Newport ST45 isolates were indistinguishable by cgMLST (Figure 1). Isolates belonging to *S*. Saintpaul ST50 appeared in two subclusters, with one major subcluster containing 18 isolates, and one minor subcluster formed by two isolates (subcluster 3 and subcluster 4 in Figure 1). Two further *S*. Saintpaul isolates were observed at distinct positions in the tree (Figure 1). Likewise, the *S*. Stanley ST29 isolates formed two clusters consisting of four and two isolates, respectively (subcluster 5 and subcluster 6 in Figure 1). Finally, the *S*. Wandsworth ST1438 isolates comprised two subclusters containing five and four isolates, respectively (subcluster 7 and subcluster 8), and two distinctly separated isolates (Figure 1).

### 3.5. Antimicrobial Resistance Genotypes 

Analysis of WGS data identified a number of antimicrobial resistance genes, including genes conferring resistance to aminoglycosides (*aadA16*, *ant(3″)-Ib, aph(3″)-Ib, aph(3′)-Ia*, *ant(3″)-IIa, aac(6′)-Ib-cr6, aph(6′)-Id)*, beta-lactams (*bla*_DHA-1_, *bla*_IND-6_, *bla*_OXA-1_, *bla*_OXA-10_, *bla*_TEM-1_, *bla*_TEM-60_), chloramphenicol (*catB3*, *catI*, *catII*), diaminopyrimidines (*dfrA14, dfrA27, dfrB4*), fluoroquinolones (*aac(6′)-Ib-cr6, qnrB4, qnrD1, qnrS1, qnrS2*), lincosamides (*linG*), macrolides (*mphA*), rifamycin (*arr-2*, *arr-3*), sulfonamides (*sul1*, *sul2*), tetracycline (*tet*(A), *tet*(D), and vancomycin (*vanX*). The distribution of antimicrobial resistance genes among the isolates is shown in Table 2. 

The *dfrA27* gene was identified in *S*. Wandsworth ST1498, while *bla*_OXA-1_, *bla*_DHA-1_, *catB3*, and *qnrB4* were found exclusively among *S*. Newport ST45 (Table 2). 

The metallo-beta-lactamase gene *bla*_IND-6_ was found in one *S.* Wandsworth ST1498 (isolate F33) (Table 2). Notably, *vanX* was identified in one *S*. Wandsworth ST1498 (isolate F86), but was not related to vancomycin resistance, because Gram-negative bacteria are insensitive to vancomycin [31]. All isolates carried either *aac(6′)-Iy* or *aac(6′)-Iaa*, which are cryptic genes that do not confer phenotypic aminoglycoside resistance [29,30]. Furthermore, all isolates contained a large number of genes encoding MDR efflux pumps, and genes encoding for antibiotic target alterations (Appendix A): among the latter, the *gyrA* (S83F) mutation was detected in one *S*. Newport ST46 (isolate F77) (Appendix A). Additional data are provided in Appendix A.

### 3.6. Antimicrobial Susceptibility Profiles

Among the 67 *Salmonella* isolates, resistance to ampicillin (*n =* 16; 24%), amoxycillin/clavulanic acid (*n =* 6; 9%), cefazolin (*n =* 6; 9%), nalidixic acid (*n =* 24; 36%), ciprofloxacin (*n =* 15; 22%), gentamicin (*n =* 1; 1.5%), kanamycin (*n =* 7; 10%), streptomycin (*n =* 9; 13%), sulfamethoxazole–trimethoprim (*n =* 8; 12%), azithromycin (*n =* 5; 7%), tetracycline (*n =* 38; 57%), and chloramphenicol (*n =* 7;10%) was observed (Table 2). MDR was observed for 14 (21%) of the isolates (Table 2). The antimicrobial susceptibility profiles are detailed in Appendix A. Notably, intermediate resistance to cefotaxime (*n =* 1; 1.5%), cefepime (*n =* 1; 1.5%), nalidixic acid (*n =* 17; 25%), ciprofloxacin (*n =* 52; 78%), streptomycin (*n =* 2; 3%), nitrofurantoin (*n =* 1; 1.5%), and chloramphenicol (*n =* 2; 3%) was found (Appendix A). The MIC values for ertapenem, imipenem, and meropenem in Wandsworth ST1498 (isolate F33) were 0.002 µg/mL, 0.094 µg/mL, and 0.08 µg/mL, respectively.

None of the isolates were resistant to cefotaxime, cefepime, fosfomycin, or nitrofurantoin (Table 2 and Appendix A). There was a correlation between phenotypic antimicrobial resistance and the presence of at least one ARG encoding resistance to aminoglycosides, amoxicillin/clavulanic acid, folate pathway inhibitors, phenicols, macrolides, and tetracycline, but no concordance between phenotypic and genotypic resistance to extended-spectrum beta-lactams and nalidixic acid (Table 2).

### 3.7. Virulence Factors

The results of the VFDB-based virulence profiling for the 67 *Salmonella* isolates are shown in Appendix A. The majority of the genes (352/496) were present universally in all isolates. The median aggregate VF scores (and ranges) for isolates belonging to different serotypes were the following: *S*. Bareilly VF 394 (393–395); *S.* Braenderup VF 409 (409–409); *S*. Hvittingfoss VF 419 (419–419); *S*. Newport VF 415 (403–433); *S*. Saintpaul VF 429 (427–434); *S*. Stanley VF 401(401–404); and *S.* Wandsworth VF 413 (408–418). Among the fimbrial adherence determinants, the majority (19/32) were common to all *Salmonella* isolates, except *cfaA, cfaB*, and *cfaC*, which were additionally present in *S*. Bareilly (*n =* 5), *S*. Stanley (*n =* 4), *S*. Wandsworth (*n =* 2), and *S*. Newport ST46 (*n =* 1). Likewise, the genes belonging to the *Salmonella enterica* type III secretion system, *sopD, sopE2, slrP, sifA, sseL, sptP, sipA/sspA, steC, steA,* and *sopD*, were common to all isolates (Appendix A). By contrast, VFs belonging to the *Aeromonas hydrophila* type VI secretion system were present solely in *S.* Wandsworth (*n =* 11) [32,33] (Appendix A). Another set of VFs that were present in all isolates were the toxin genes *cyaB*, *cyaD*, *senB*, *cylA*, and *plcD* [34].

## 4. Discussion

Although frog meat is a popular food worldwide [35], there is a lack of information regarding the occurrence of *Salmonella* in frogs sold for human consumption. In the current study, the overall prevalence of *Salmonella* carriage among *H. rugulosus* obtained from wet markets in Hong Kong was 65%, and included a variety of *Salmonella* serotypes, with many isolates featuring genotypic and phenotypic resistance to clinically relevant antimicrobials.

Among the isolates, several serovars, including *S.* Hvittingfoss, *S.* Newport, *S.* Stanley, and *S.* Wandsworth, had been described previously in *H. rugulosus* farmed in Thailand [36]. Furthermore, *S*. Saintpaul and *S*. Wandsworth had been isolated from *H. rugulosus* used as feeder frogs for captive reptiles in Thailand [37]. These reports are suggestive of frogs as natural hosts for these serotypes. Some serotypes in this study are infrequently reported, but have nevertheless been associated with human cases of salmonellosis. For example, the rare serotype *S.* Hvittingfoss ST446, found in three frogs in this study, was recently linked to an outbreak of salmonellosis in travellers returning from Hong Kong [38]. Likewise, *S*. Wandsworth, identified in 11 frogs, was associated with infections in small children in the USA in 2009, and with peritonitis in a patient in Hong Kong in 2022 [39,40]. The identification of this rare serotype among 16% of the frogs indicates that *S*. Wandsworth may be widespread among frogs, and that frogs may represent an important reservoir for this serotype. Other serotypes found in this study are more frequent among human infections, and have also been implicated in recent disease outbreaks in the EU (*S*. Bareilly, *S*. Braenderup, *S*. Newport, *S*. Stanley) [6], or are endemic in Asia but less frequent in Europe, such as *S*. Stanley [41]. 

Phylogenetic analysis showed that many of the *Salmonella* STs in this study were closely related by cgMLST, with subclusters observed among isolates from frogs obtained from different markets. Despite the lack of details on the acquisition of the frogs sold at the wet markets, these subclusters may be considered to reflect a common origin of the frogs, e.g., farm or location of capture. In particular, the clonality of the MDR *S*. Newport ST45 isolates is suggestive of a lineage deriving from a common source. 

Multiple genes conferring resistance to antimicrobials that are listed as critically important, or very important to human health, were detected [42]. One of the clinically relevant AMR genes included *bla*_DHA-1_, identified among six MDR *S*. Newport ST45. DHA-1 belongs to the family of plasmid-mediated AmpC beta-lactamases that confer resistance to the beta-lactamase inhibitor clavulanic acid [43]. Accordingly, all DHA-1-producing *S*. Newport in this study were phenotypically resistant to ampicillin and to amoxycillin/clavulanic acid. The *bla*_DHA-1_ gene is infrequently reported in *Salmonella* and other Enterobacterales [44]. To the best of our knowledge, this is the first report of *bla*_DHA-1_ in *S*. Newport isolated from amphibians, although one DHA-1-producing human clinical *S*. Newport was described in Spain between 2004 and 2009 [45]. One of the most prevalent AMR genes in this study was *bla*_TEM-60,_ which encodes an extended-spectrum beta-lactamase (ESBL); however, in this study, *bla*_TEM-60_-carriers failed to exhibit a detectable ESBL phenotype. Likewise, the *bla*_IND-6_-harbouring *S*. Wandsworth did not exhibit resistance to carbapenems. By contrast, the abundance of the plasmid-mediated quinolone resistance (PMQR) genes *aac(6′)-Ib-cr6*, *qnrB4*, *qnrS1*, and *qnrS2* among the isolates was reflected in a high proportion of nalidixic-acid- and ciprofloxacin-resistant isolates. Markedly, none of the isolates in this study were fully susceptible to ciprofloxacin. PMQR among *Salmonella* is of concern, because it has the potential to spread horizontally, and, although PMQR genes only confer low-level resistance to quinolones, they promote step-wise development to high-level fluoroquinolone resistance [46,47]. The presence of PMQR genes in human pathogens isolated from animals destined for human consumption is of concern, because *Salmonella* with decreased ciprofloxacin resistance carrying PMQR genes have been linked to outbreaks, as previously described, in the UK with *S*. Virchow [48], in the USA with *S*. Newport [49], and in Spain with *S*. Corvallis [50].

With resistance to fluoroquinolones increasing worldwide, azithromycin is currently considered an important alternative for the treatment of invasive salmonellosis [42]. Azithromycin resistance in human NTS is currently rare in the USA (0.5%) and Europe (0.8%) [51,52]. Moreover, resistance to azithromycin was either not detected, or was identified only at very low levels in food-producing animals, including calves (0%), broilers (1.9%), pigs (0.3%), and turkeys (1.2%), in the EU, between 2019 and 2020 [52]. By contrast, azithromycin resistance, mainly associated with the presence of *mphA*, has been increasing over time, and seems to be more prevalent in human NTS in Europe than in Taiwan (3.1%) [51], and in *Salmonella* from humans, animals, and food in China (3.9%) [53]. The observation, in our study, that azithromycin resistance and the *mph* gene occur more frequently in *Salmonella* of amphibian origin (5.7%) than in *Salmonella* of livestock or human origin may suggest the potential of *H. rugulosus* as a source of dissemination of azithromycin-resistant *Salmonella*.

Animals including amphibians may acquire antimicrobial-resistant bacteria from other animals, their environment, or their feed. Little information is available regarding frog farming and the usage of antimicrobial agents in frog farms in Asia; however, a recent report from Vietnam indicates that antimicrobials, such as amoxicillin, enrofloxacin, ciprofloxacin, sulfamethoxazole–trimethoprim, doxycycline, oxytetracycline, tetracycline, florfenicol, and rifampicin, are commonly applied empirically by frog farmers to treat frog diseases [16,54]: thus, the use and misuse of large amounts of different antimicrobials may select for resistant strains in farmed frogs. Although information on the origins of the frogs analysed in this study was available at country level, data on the aquacultural or natural settings were lacking, and it cannot be excluded that many of the frogs may have been farmed under comparable conditions. Similarly to our study, a recent investigation on wet markets in Hong Kong recently reported a high prevalence (42%) of *Salmonella*, and a high proportion of AMR isolates, among edible freshwater turtles, further underlining the potential risk of *Salmonella* and AMR in non-traditional aquaculture [55].

The presence of genotypic and phenotypic resistance characteristics in *Salmonella* in food animals is worrying, and indicates that first-line antibiotic ciprofloxacin, as well as ampicillin, amoxycillin/clavulanic acid, and azithromycin, used against such strains, may fail, should infectious transmissions to humans occur. Likewise, isolates showing intermediate resistance, i.e., reduced susceptibility, are of concern for public health. Reduced susceptibility is important, because it facilitates the selection of isolates with higher-level resistance, and contributes to the development and spread of resistance to critically important antimicrobial agents, such as ciprofloxacin [46,47]. 

The in silico screening of the VFs revealed an abundance of VFs in all the isolates whilst, concurrently, the differences in VF content among the various serotypes was rather unremarkable: these findings are in agreement with previous studies, in which VFs were reportedly conserved among NT *Salmonella* serotypes [56]. Based on the distribution of the VFs, and the lack of variation of median aggregate VF scores among the serotypes, there was no specific *Salmonella* serotype in this study that showed a distinct virulence profile; however, the universal presence of fimbrial genes, genes encoding secretion systems, and toxins among the isolates was an indication that these *Salmonella* serotypes have the potential to cause disease in humans.

This study had some limitations. Firstly, the interpretation of the results should take into consideration that sampling was restricted to *H. rugulosus* because this is the only frog species sold in wet markets in Hong Kong: thus, the results cannot be generalised to other frog species that may be available for human consumption elsewhere. Secondly, there were unequal sample sizes among the different wet markets from which the frogs were obtained; therefore, there remains the possibility of unintended overrepresentation of the proportion of positive samples from individual markets, which were randomly selected. Thirdly, the study was challenged by the lack of any previous data on the prevalence of *Salmonella* in frogs, and lack of data on frog import and retail volumes in Hong Kong; therefore, essentially, this study was a pilot study, to collect some baseline data. Finally, the sampling period was short: it should be considered that the prevalence of *Salmonella* among the frogs may have been subject to unknown factors—for example, fluctuations among the suppliers of the wet markets—and that such differences would have remained undetected.

## 5. Conclusions

This study demonstrates that a high proportion of live frogs sold for human consumption in wet markets are carriers of *Salmonella*, including serotypes that are frequently linked to cases of human disease, and *Salmonella* that contain clinically relevant AMR genes. To the best of the authors’ knowledge, no official advice or guidelines exist regarding the handling of live edible frogs. To this end, the results presented in this study offer useful information for specialists in the public health and food safety sectors. Raised awareness and public health recommendations for hygiene requirements at the stages of handling and processing of edible frogs and their meat should be considered, to mitigate the risk of transmission of susceptible and resistant *Salmonella* to humans. The role of edible frogs and derived meat products, e.g., frog legs, as sources of antimicrobial-resistant and susceptible *Salmonella* and other foodborne pathogens, needs to be further investigated in future. 

## Figures and Tables

**Figure 1 foods-12-02245-f001:**
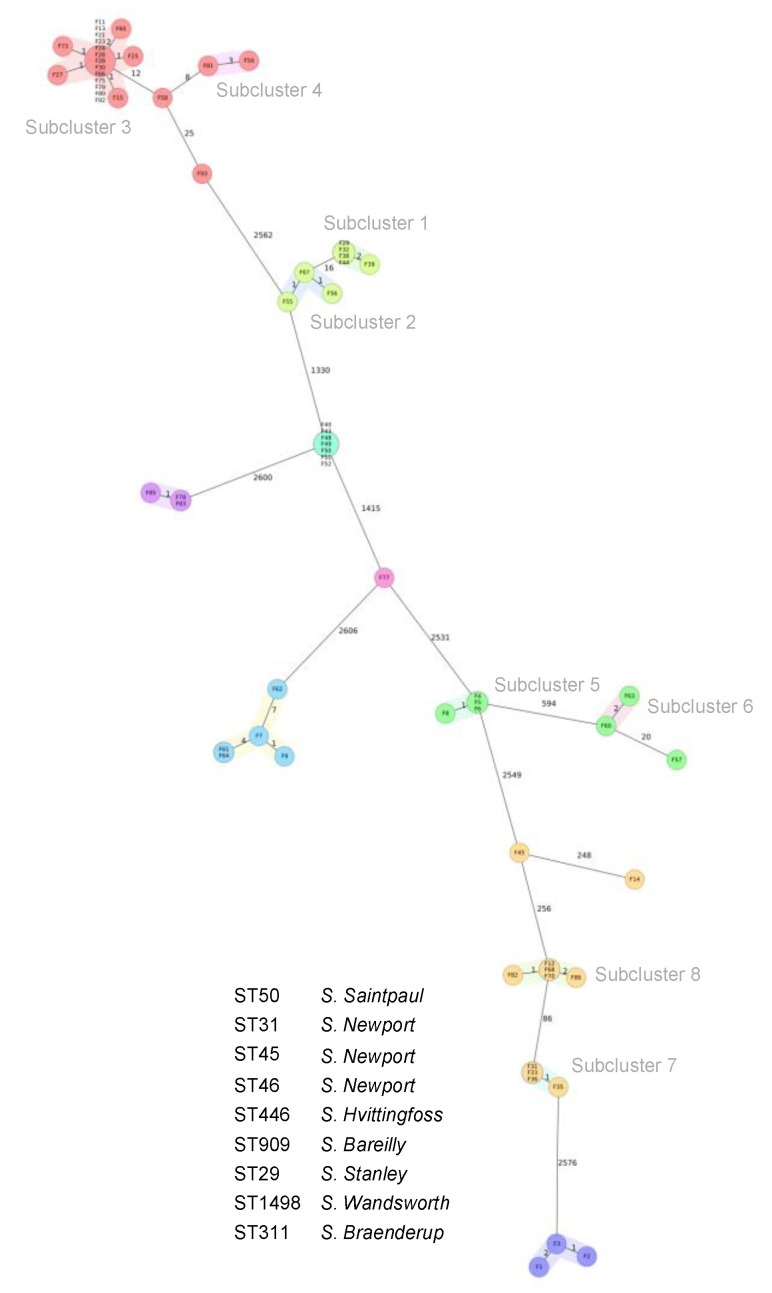
Phylogenetic relationship of 67 *Salmonella* serotypes isolated from Chinese edible frogs (*Hoplobatrachus rugulosus*), based on their core genome multilocus sequence type (cgMLST) allelic profiles. The minimum spanning tree was generated using SeqSphere+ (Ridom GmbH). The colours of the circles represent STs according to in silico MLST. *Salmonella* serotypes and subclusters 1–8 are indicated. Clusters are displayed by pastel background colours. Numbers on connecting lines indicate the number of allelic differences between two strains. Each circle contains the strain ID(s), as shown in Table 2.

**Table 1 foods-12-02245-t001:** Origin, number of samples, and proportion of *Salmonella*-positive samples from Chinese edible frogs (*Hoplobatrachus rugulosus*) from nine different wet markets in Hong Kong.

Wet Market	Samples (*n*)	Positive Samples (*n*)	Proportion of Positive Samples (%)	*Salmonella* Serotypes Identified (*n*)
A	20	16	80	*S*. Newport (4), *S*. Saintpaul (8), *S*. Wandsworth (4)
B	21	15	71	*S.* Hvittingfoss (3), *S*. Newport (2), *S*. Saintpaul (7), *S*. Wandsworth (3)
C	18	3	17	*S*. Saintpaul (2), *S*. Wandsworth (1)
D	6	6	100	*S*. Bareilly (2), *S*. Stanley (4)
E	10	10	100	*S*. Bareilly (3), *S*. Newport (2), *S*. Saintpaul (2), *S*. Stanley (3)
F	10	5	50	*S*. Saintpaul (3), *S*. Wandsworth (2)
G	10	6	60	*S*. Newport (5), *S*. Wandsworth (1)
H	3	3	100	*S*. Braenderup (3)
I	5	3	60	*S*. Newport (3)

**Table 2 foods-12-02245-t002:** Genotypic and phenotypic characteristics of *Salmonella* isolated from Chinese edible frogs (*Hoplobatrachus rugulosus*).

Isolate ID	Serotype	ST	Antimicrobial Resistance Genes ^a^	Resistance Profile ^b^	GenBank Accession No.
F7	*S.* Bareilly	909	*aac(6′)-Iy*, *ant(3″)-Ib*, *ant(3″)-Iia*, *bla*_TEM-1_, *bla*_TEM-60_, *linG*, *mphA*, *mrX, qnrS1, tet*(A)	Beta-lactams (AM), macrolides (AZM), aminoglycosides (S), tetracyclines (TE)	JASBEW000000000
F9	*S.* Bareilly	909	*aac(6′)-Iy*, *ant(3″)-Ib*, *ant(3″)-Iia*, *bla*_TEM-1_, *bla*_TEM-60_, *linG*, *mphA*, *mrX, qnrS1, tet*(A)	Beta-lactams (AM), macrolides (AZM), aminoglycosides (S), tetracyclines (TE)	JASBEI000000000
F61	*S.* Bareilly	909	*aac(6′)-Iy*, *ant(3″)-Ib*, *ant(3″)-Iia*, *bla*_TEM-1_, *bla*_TEM-60_, *linG*, *mphA*, *mrX, qnrS1, tet*(A)	Beta-lactams (AM), macrolides (AZM), aminoglycosides (S), tetracyclines (TE)	JASBFI000000000
F62	*S.* Bareilly	909	*aac(6′)-Iy*, *bla*_TEM-60_	–	JASBFD000000000
F64	*S.* Bareilly	909	*aac(6′)-Iy*, *ant(3″)-Ib*, *ant(3″)-Iia*, *bla*_TEM-1_, *bla*_TEM-60_, *linG*, *mphA*, *mrX, qnrS1, tet*(A)	Beta-lactams (AM), macrolides (AZM), aminoglycosides (S), tetracyclines (TE)	JASBFG000000000
F1	*S.* Braenderup	311	*aac(6′)-Iaa*, *bla*_TEM-60_, *qnrS2*	–	JASBGW000000000
F2	*S.* Braenderup	311	*aac(6′)-Iaa*, *bla*_TEM-60_	Aminoglycosides (S)	JASBGV000000000
F3	*S.* Braenderup	311	*aac(6′)-Iaa*, *bla*_TEM-60_, *qnrS2*	Quinolones (NA)	JASBGH000000000
F78	*S.* Hvittingfoss	446	*aac(6′)-Iy*, *bla*_TEM-60_, *qnrS2, tet*(A)	Fluoroquinolones (CIP), tetracyclines (TE)	JASBET000000000
F83	*S.* Hvittingfoss	446	*aac(6′)-Iy*, *bla*_TEM-60_, *qnrS2, tet*(A)	tetracyclines (TE)	JASBEL000000000
F85	*S.* Hvittingfoss	446	*aac(6′)-Iy*, *bla*_TEM-60_, *qnrS2, tet*(A)	tetracyclines (TE)	JASBEN000000000
F29	*S.* Newport	31	*aac(6′)-Iy*, *bla*_TEM-1_, *bla*_TEM-60_, *qnrS1, tet*(A)	Beta-lactams (AM), tetracyclines (TE)	JASBGL000000000
F32	*S.* Newport	31	*aac(6′)-Iy*, *bla*_TEM-1_, *bla*_TEM-60_, *qnrS1, tet*(A)	Beta-lactams (AM), tetracyclines (TE)	JASBGC000000000
F38	*S.* Newport	31	*aac(6′)-Iy*, *bla*_TEM-60_	–	JASBGA000000000
F39	*S.* Newport	31	*aac(6′)-Iy*, *bla*_TEM-60_, *bla*_TEM-1_, *qnrS1, tet*(A)	Beta-lactams (AM), tetracyclines (TE)	JASBFZ000000000
F44	*S.* Newport	31	*aac(6′)-Iy*, *bla*_TEM-1_, *bla*_TEM-60_, *qnrS1, tet*(A)	Beta-lactams (AM), tetracyclines (TE)	JASBFW000000000
F55	*S.* Newport	31	*aac(6′)-Iy*, *bla*_TEM-60_, *dfrA14*, *qnrS1, tet*(A)	Sulfonamides (SXT), tetracyclines (TE)	JASBFO000000000
F56	*S.* Newport	31	*aac(6′)-Iy*, *bla*_TEM-60_, *dfrA14*, *qnrS1, tet*(A)	Sulfonamides (SXT), tetracyclines (TE)	JASBFN000000000
F67	*S.* Newport	31	*aac(6′)-Iy*, *bla*_TEM-60_, *dfrA14*, *qnrS1, tet*(A)	Fluoroquinolones (CIP), sulfonamides (SXT), tetracyclines (TE)	JASBFB000000000
F40	*S.* Newport	45	*aac(6′)-Ib-cr6*, *aac(6′)-Iy*, *aph(3″)-Ib*, *aph(3′)-Ia*, *aph(6)-Id*, *arr-3*, *bla*_DHA-1_, *bla*_OXA-1_, *catB3*, *catII*, *qnrB4*, *sul1*, *sul2, tet*(A), *tet*(D)	Beta-lactams (AM, AMC, CZ), amphenicols (C), fluoroquinolones (CIP), aminoglycosides (K, S), tetracyclines (TE)	JASBFX000000000
F43	*S.* Newport	45	*aac(6′)-Ib-cr6*, *aac(6′)-Iy*, *aph(3″)-Ib*, *aph(3′)-Ia*, *aph(6)-Id*, *arr-3*, *bla*_DHA-1_, *bla*_OXA-1_, *catB3*, *catII*, *qnrB4*, *sul1*, *sul2*, *tet*(D)	Beta-lactams (AM, AMC, CZ), amphenicols (C), aminoglycosides (K, S), tetracyclines (TE)	JASBFV000000000
F48	*S.* Newport	45	*aac(6′)-Ib-cr6*, *aac(6′)-Iy*, *aph(3″)-Ib*, *aph(3′)-Ia*, *aph(6)-Id*, *arr-3*, *bla*_OXA-1_, *catB3*, *catII*, *sul1*, *sul2, tet*(A), *tet*(D)	Beta-lactams (AM), amphenicols (C), fluoroquinolones (CIP), aminoglycosides (K), tetracyclines (TE)	JASBFU000000000
F49	*S.* Newport	45	*aac(6′)-Ib-cr6*, *aac(6′)-Iy*, *aph(3″)-Ib*, *aph(6)-Id*, *arr-3*, *bla*_DHA-1_, *bla*_OXA-1_, *catB3*, *catII*, *qnrB4*, *sul1*, *sul2, tet*(A), *tet*(D)	Beta-lactams (AM, AMC, CZ), amphenicols (C), aminoglycosides (K, S), tetracyclines (TE)	JASBFR000000000
F50	*S.* Newport	45	*aac(6′)-Ib-cr6*, *aac(6′)-Iy*, *aph(3″)-Ib*, *aph(3′)-Ia*, *aph(6)-Id*, *arr-3*, *bla*_DHA-1_, *bla*_OXA-1_, *catB3*, *catII*, *qnrB4*, *sul1*, *sul2, tet*(A), *tet*(D)	Beta-lactams (AM, AMC, CZ), amphenicols (C), aminoglycosides (K, S), tetracyclines (TE)	JASBFQ000000000
F51	*S.* Newport	45	*aac(6′)-Ib-cr6*, *aac(6′)-Iy*, *aph(3″)-Ib*, *aph(3′)-Ia*, *aph(6)-Id*, *arr-3*, *bla*_DHA-1_, *bla*_OXA-1_, *catB3*, *catII*, *qnrB4*, *sul1*, *sul2, tet*(A), *tet*(D)	Beta-lactams (AM, AMC, CZ), amphenicols (C), fluoroquinolones (CIP), aminoglycosides (GM, K, S), tetracyclines (TE)	JASBFL000000000
F52	*S.* Newport	45	*aac(6′)-Ib-cr6*, *aac(6′)-Iy*, *aph(3″)-Ib*, *aph(6)-Id*, *arr-3*, *bla*_DHA-1_, *bla*_OXA-1_, *catB3*, *catII*, *qnrB4*, *sul1*, *sul2, tet*(A), *tet*(D)	Beta-lactams (AM, AMC, CZ), amphenicols (C), fluoroquinolones (CIP), aminoglycosides (K, S), tetracyclines (TE)	JASBFP000000000
F77	*S.* Newport	46	*aac(6′)-Iy, tet*(A)	Quinolones (NA), tetracyclines (TE)	JASBEQ000000000
F11	*S.* Saintpaul	50	*aac(6′)-Iy*, *bla*_TEM-60_, *qnrS1, tet*(A)	(Fluoro)quinolones (CIP, NA), tetracyclines (TE)	JASBGS000000000
F13	*S.* Saintpaul	50	*aac(6′)-Iy*, *bla*_TEM-60_	Quinolones (NA)	JASBGR000000000
F15	*S.* Saintpaul	50	*aac(6′)-Iy*, *bla*_TEM-60_, *qnrS1, tet*(A)	Quinolones (NA), tetracyclines (TE)	JASBGT000000000
F21	*S.* Saintpaul	50	*aac(6′)-Iy*, *bla*_TEM-60_	Quinolones (NA)	JASBGP000000000
F23	*S.* Saintpaul	50	*aac(6′)-Iy*, *bla*_TEM-60_	Quinolones (NA)	JASBGO000000000
F24	*S.* Saintpaul	50	*aac(6′)-Iy*, *bla*_TEM-60_	Quinolones (NA)	JASBGJ000000000
F25	*S.* Saintpaul	50	*aac(6′)-Iy*, *bla*_TEM-60_	Quinolones (NA)	JASBGN000000000
F26	*S.* Saintpaul	50	*aac(6′)-Iy*, *bla*_TEM-60_	Quinolones (NA)	JASBGI000000000
F27	*S.* Saintpaul	50	*aac(6′)-Iy*, *bla*_TEM-60_	Quinolones (NA)	JASBGK000000000
F28	*S.* Saintpaul	50	*aac(6′)-Iy*, *bla*_TEM-60_, *qnrS1, tet*(A)	(Fluoro)quinolones (CIP, NA), tetracyclines (TE)	JASBGM000000000
F30	*S.* Saintpaul	50	*aac(6′)-Iy*, *bla*_TEM-60_	Quinolones (NA)	JASBGD000000000
F65	*S.* Saintpaul	50	*aac(6′)-Iy*, *bla*_TEM-60_, *qnrS1, tet*(A)	(Fluoro)quinolones (CIP, NA), tetracyclines (TE)	JASBFC000000000
F66	*S.* Saintpaul	50	*aac(6′)-Iy*, *bla*_TEM-60_	Quinolones (NA)	JASBEZ000000000
F73	*S.* Saintpaul	50	*aac(6′)-Iy*, *bla*_TEM-60_, *qnrS1, tet*(A)	(Fluoro)quinolones (CIP, NA), tetracyclines (TE)	JASBEY000000000
F75	*S.* Saintpaul	50	*aac(6′)-Iy*, *bla*_TEM-60_, *qnrS1, tet*(A)	(Fluoro)quinolones (CIP, NA), tetracyclines (TE)	JASBEP000000000
F79	*S.* Saintpaul	50	*aac(6′)-Iy*, *bla*_TEM-60_, *qnrS1, tet*(A)	(Fluoro)quinolones (CIP, NA), tetracyclines (TE)	JASBEU000000000
F80	*S.* Saintpaul	50	*aac(6′)-Iy*, *bla*_TEM-60_	Quinolones (NA)	JASBEV000000000
F92	*S.* Saintpaul	50	*aac(6′)-Iy*, *bla*_TEM-60_, *qnrS1, tet*(A)	(Fluoro)quinolones (CIP, NA), tetracyclines (TE)	JASBEJ000000000
F59	*S.* Saintpaul	50	*aac(6′)-Iy*, *bla*_TEM-60_, *qnrS1, tet*(A)	Quinolones (NA), tetracyclines (TE)	JASBFE000000000
F81	*S.* Saintpaul	50	*aac(6′)-Iy*, *bla*_TEM-60_	Quinolones (NA)	JASBES000000000
F58	*S.* Saintpaul	50	*aac(6′)-Iy*, *bla*_TEM-60_	Quinolones (NA)	JASBFK000000000
F93	*S.* Saintpaul	50	*aac(6′)-Iy*, *ant(3″)-IIa*, *ant(3″)-IIa*, *arr-2*, *bla*_OXA-10_, *bla*_TEM-60_, *catI*, *dfrA14*, *dfrB4*, *mphA*, *mrX, qnrS1*, *sul1*, *sul2, tet*(A)	Beta-lactams (AM), macrolides (AZM), (fluoro)quinolones (CIP, NA), tetracyclines (TE)	JASBEK000000000
F4	*S.* Stanley	29	*aac(6′)-Iy*, *bla*_TEM-60_	–	JASBFT000000000
F5	*S.* Stanley	29	*aac(6′)-Iy*, *bla*_TEM-60_	–	JASBFS000000000
F6	*S.* Stanley	29	*aac(6′)-Iy*, *bla*_TEM-60_	–	JASBFF000000000
F8	*S.* Stanley	29	*aac(6′)-Iy*, *bla*_TEM-60_	–	JASBEO000000000
F60	*S.* Stanley	29	*aac(6′)-Iy*, *bla*_TEM-60_	–	JASBFH000000000
F63	*S.* Stanley	29	*aac(6′)-Iy*, *bla*_TEM-60_	–	JASBFJ000000000
F57	*S.* Stanley	29	*aac(6′)-Iy*, *bla*_TEM-60_	–	JASBFM000000000
F12	*S.* Wandsworth	1498	*aac(6′)-Iy*, *bla*_TEM-60_, *qnrS2*	–	JASBGQ000000000
F68	*S.* Wandsworth	1498	*aac(6′)-Iy*, *bla*_TEM-60_, *qnrS2*	–	JASBFA000000000
F70	*S.* Wandsworth	1498	*aac(6′)-Iy*, *bla*_TEM-60_, *qnrS2*	–	JASBEX000000000
F82	*S.* Wandsworth	1498	*aac(6′)-Ib-cr6*, *aac(6′)-Iy*, *aadA16*, *arr-3*, *bla*_TEM-60_, *dfrA27*, *qnrS2*, *sul1, tet*(A)	Fluoroquinolones (CIP), sulfonamides (SXT), tetracyclines (TE)	JASBER000000000
F86	*S.* Wandsworth	1498	*aac(6′)-Iy*, *bla*_TEM-60_, *qnrS2, vanX*	–	JASBEM000000000
F31	*S.* Wandsworth	1498	*aac(6′)-Ib-cr6*, *aac(6′)-Iy*, *aadA16*, *ant(3″)-IIa*, *arr-3*, *bla*_TEM-60_, *dfrA27*, *linG*, *qnrS1*, *sul1, tet*(A)	Sulfonamides (SXT), tetracyclines (TE)	JASBGB000000000
F33	*S.* Wandsworth	1498	*aac(6′)-Ib-cr6*, *aac(6′)-Iy*, *aadA16*, *ant(3″)-IIa*, *arr-3*, *bla_IND-6_, bla*_TEM-60_, *dfrA27*, *linG*, *qnrS1*, *sul1, tet*(A)	Sulfonamides (SXT), tetracyclines (TE)	JASBGG000000000
F35	*S.* Wandsworth	1498	*aac(6′)-Ib-cr6*, *aac(6′)-Iy*, *aadA16*, *ant(3″)-IIa*, *arr-3*, *bla*_TEM-60_, *dfrA27*, *linG*, *qnrS1*, *sul1, tet*(A)	Sulfonamides (SXT), tetracyclines (TE)	JASBGE000000000
F36	*S.* Wandsworth	1498	*aac(6′)-Ib-cr6*, *aac(6′)-Iy*, *aadA16*, *ant(3″)-IIa*, *arr-3*, *bla*_TEM-60_, *dfrA27*, *linG*, *qnrS1*, *sul1, tet*(A)	Sulfonamides (SXT), tetracyclines (TE)	JASBGF000000000
F14	*S.* Wandsworth	1498	*aac(6′)-Iy*, *bla*_TEM-60_, *qnrS1, tet*(A)	Tetracyclines (TE)	JASBGU000000000
F45	*S.* Wandsworth	1498	*aac(6′)-Iy*, *bla*_TEM-60_, *qnrD1*	–	JASBFY000000000

^a^ *aac(6′)-Iy* and *aac(6′)-Iaa* are cryptic, and do not confer phenotypic aminoglycoside resistance [29,30]. ^b^ Resistance profiles show the antimicrobial class and, in brackets, the individual antimicrobials. AM: ampicillin; AMC: amoxicillin/clavulanic acid; AZM: azithromycin; CIP: ciprofloxacin; CZ: cefazoline; GM: gentamicin; K: kanamycin; NA: nalidixic acid; ST: sequence type; SXT: sulfamethoxazole-trimethoprim; TE: tetracycline.

## Data Availability

Sequencing read data and genome assemblies have been deposited under BioProject accession number PRJNA966132. Accession numbers for the individual isolates from this study are listed in Table 2.

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
