# Peer review of "Serotypes, Antimicrobial Resistance Profiles, and Virulence Factors of Salmonella Isolates in Chinese Edible Frogs (Hoplobatrachus rugulosus) Collected from Wet Markets in Hong Kong"

_foods, 2023, doi:10.3390/foods12112245_

Round 1
Reviewer 1 Report
The manuscript entitled “Phenotypic and genotypic characteristics of Salmonella isolates in Chinese edible frogs (Hoplobatrachus rugulosus) collected from wet markets in Hong Kong” is a good work and such studies are rare. However, the following issues are to be addressed,
Major concerns
Ø As the authors have done whole genome sequencing of all the recovered Salmonella isolates, it is suggested to undertake virulence profiling (genes with virulence potential) using genomic features.
Ø I think the authors have used ISO 6579-1: 2017 method for isolation and identification of Salmonella. It should be cited. The authors should also clarify whether any other plating media other than XLD was used to confirm atypical colonies. Any biochemical characterization done?
Ø In MALDI-TOF, which bacterial strain was inoculated on the calibration spots as a calibration and internal identification control.
Minor
Abstract: Please mention about phylogenetic relationship of 67 Salmonella isolated from different locations
MAR index may be computed and added to the tables
Line 28: …..human consumption in wet markets are carriers of multidrug resistant Salmonella
Line 29 and 354: “Public health recommendations for handling edible frogs”. Are there any such recommendations. Please discuss.
Line 135: Antimicrobial agents
Line 165: Frogs containing Salmonella……… Reword
Line 321: “Animals including amphibians may acquire AMR genes” Rephrase.. It is bacteria which acquires AMR genes
Minor editing required
Author Response
Authors' point-by-point replies to
REVIEWER #1:
The manuscript entitled “Phenotypic and genotypic characteristics of Salmonella isolates in Chinese edible frogs (Hoplobatrachus rugulosus) collected from wet markets in Hong Kong” is a good work and such studies are rare. However, the following issues are to be addressed,
Major concerns
Ø As the authors have done whole genome sequencing of all the recovered Salmonella isolates, it is suggested to undertake virulence profiling (genes with virulence potential) using genomic features
REPLY:
Thank you for this suggestion. We have performed virulence profiling for all 67 isolates. Please find below the amendments to the manuscript:
Line24: ...and a high number of virulence determinants were identified.
Line 79: virulence profling,...
Line 145: Virulence factors (VFs) were identified by a bi-directional best-hit approach using Diamond with standard setting [25]. The Prokka-predicted proteomes were compared to the representative-proteins data set A of the virulence factor database VFDB, downloaded in Mai 2023 [26]. The aggregate VF score was defined as the number of unique VF detected for each isolate
Line 283:
3.7 Virulence factors
The results of the VFDB-based virulence profiling for the 67 Salmonella isolates are shown in supplemental Table S3. The majority of the genes (352/496) were present universally in all isolates. The median aggregate VF scores (and ranges) for isolates belonging to different serotypes were the following: S. Bareilly VF 394 (393–395), S. Braenderup VF 409 (409–409), S. Hvittingfoss VF 419 (419–419), S. Newport VF 415 (403–433), S. Saintpaul VF 429 (427–434),S. Stanley VF 401(401–404), and S. Wandsworth VF 413 (408–418) Among the fimbrial adherence determinants, the majority (19/32) were common to all Salmonella isolates, except for cfaA, cfaB and cfaC which were additionally present in S. Bareilly (n=5), S. Stanley (n=4), S. Wandsworth (n=2), and S. Newport ST46 (n=1). Likewise, the genes belonging to Salmonella enterica type III secretion system sopD, sopE2, slrP, sifA, sseL, sptP, sipA/sspA, steC, steA, and sopD were common to all isolates (supplemental Table S3). By constrast, VFs belonging to the Aeromonas hydrophila type VI secretion system were present solely in S. Wandsworth (n=11) [32,33] (supplemental Table S3). Another set of VFs that were present in all isolates were the toxin genes cyaB, cyaD, senB, cylA, and plcD [34].
Line 391: The in silico screening of the VFs revealed an abundance of VFs in all the isolates whilst concurrently, the differences in VF content among the various serotypes was rather unremarkable. These findings are in agreement with previous studies in which VFs were reportedly conserved among NT Salmonella serotypes [56]. Based on the distribution of the VFs and the lack of variation of median aggregate VF scores among the serotypes, there was no specific Salmonella serotype in this study that showed a distinct virulence profile. However, the universal presence of fimbrial genes, genes encoding secretion systems and toxins among the isolates are an indication that these Salmonella serotypes have the potential to cause disease in humans.
Ø I think the authors have used ISO 6579-1: 2017 method for isolation and identification of Salmonella. It should be cited. The authors should also clarify whether any other plating media other than XLD was used to confirm atypical colonies. Any biochemical characterization done?
REPLY:
Thank you for this observation. The method for isolation and identification of Salmonella used in this study corresponds to that of ISO 6579-1. An appropriate reference to the ISO 6579-1 method has been inserted in the paragraph under the subheading "2.2. Salmonella isolation and identification".
Line 104: Salmonella spp. detection was performed using the ISO 6579-1: 2017 method for isolation and identification of Salmonella [20].
In addition, it is now mentioned that no other plating media other than XLD was used, and no other biochemical characterizations were done (Line 112)
Ø In MALDI-TOF, which bacterial strain was inoculated on the calibration spots as a calibration and internal identification control.
REPLY:
This information has been added to the manuscript:
Line 116: For calibration and internal quality control of the MALDI Biotyper System, the Bruker Bacterial Test Standard (BTS) containing extract of Escherichia coli DH5 alpha was used, according to the manufacturer's instructions (Bruker, Massachusetts, US). Salmonella Typhimurium ATCC 14028 was used for internal evaluation.
Minor
Abstract: Please mention about phylogenetic relationship of 67 Salmonella isolated from different locations
REPLY:
An amendment has been made to the abstract to mention the phylogenetic relatedness of the isolates. Subsequently, the abstract has modified in order not to exceed the word count of 200:
Line 23: Many isolates were phylogenetically related.
MAR index may be computed and added to the tables
REPLY:
Thank you for this interesting suggestion. We have carefully considered this aspect but have come to the conclusion that MAR indexing would not add value to the manuscript or contribute significantly to main conclusions and highlights of this study.
Line 28: …..human consumption in wet markets are carriers of multidrug resistant Salmonella
REPLY:
The term "antimicrobial resistant" has been replaced with "multidrug" as suggested by Reviewer #3 (Line 28).
Line 29 and 354: “Public health recommendations for handling edible frogs”. Are there any such recommendations. Please discuss.
REPLY:
Thank you for this comment. As far as the authors are aware, there exist no such recommendations. An amendment has been made to the Conclusions to highlight this fact.
Line 418: To the best of the authors’ knowledge, no official advice or guidelines exist regarding the handling of live edible frogs.
Line 135: Antimicrobial agents
REPLY:
The term " substances" has been replaced with "agents" as suggested by Reviewer #3 (Line 152).
Line 165: Frogs containing Salmonella……… Reword
REPLY:
The phrase has been reworded:
Line 185 (and line 190): Frogs harbouring Salmonella...
Line 321: “Animals including amphibians may acquire AMR genes” Rephrase.. It is bacteria which acquires AMR genes
REPLY:
Thank you for this comment. The phrase has been amended as follows:
Line 369: Animals including amphibians may acquire antimicrobial resistant bacteria from other animals..."
Reviewer 2 Report
This study is assessing the occurrence of Salmonella isolates in edible Chinese frogs and reports the phenotypic and genotypic characteristics using WGS. I have a few minor comments:
Line 36 – The authors should include the most recent data on reporting how many are affected worldwide, perhaps use CDC data as the source. This cited reference is from 2010.
Line 71-73 - Is this the first report of this kind of study or are there any studies done previously? The authors have mentioned these studies as lacking, they need to cite any corelating reference if available although the authors have mentioned the presence of Salmonella in H. rugulosus in Discussion (Line 262-276) but I would suggest to cite more reference if published.
Line 94 – How many swabs per frog were collected?
Line 113 - Please provide a supplemental file with the genome assembly statistics and metadata for WGS.
Line 115 – Was it DNA Blood and Tissue kit or DNeasy kit from Qiagen?
Line 117 – Which reagent kit from Illumina did the authors use?
In Line 19, 80-82, authors mention that frogs were collected from Hong Kong markets whereas in Line 160-162 authors mention origination of frogs from mainland China and Thailand. These line lacks clarity, please clarify.
Table 1: Please mention the wet market names, city, and state as a footnote. Also, what does isolate ID mean, are all these from Farm F?
Figure 1: Please indicate the sub clusters in the list given and what sub cluster is the purple group?
Line 237 – 250 - in silico and n= should be in italics throughout the manuscript
Section 3.5 and 3.6 - It would be nice if the authors briefly presented a table of sidewise comparison of phenotype and genotype (detected resistance genes profile) results from all the salmonella isolates. If the authors add the concordance between phenotypic detected resistance and predicted resistance from genomes then, it will make a good addition.
The authors have found several ARGs in the salmonella isolates, it would be nice if the authors could add the mobile genetic elements that harbor ARG.
It is well written.
Author Response
Authors' point-by-point replies to
REVIEWER #2
This study is assessing the occurrence of Salmonella isolates in edible Chinese frogs and reports the phenotypic and genotypic characteristics using WGS. I have a few minor comments:
Line 36 – The authors should include the most recent data on reporting how many are affected worldwide, perhaps use CDC data as the source. This cited reference is from 2010.
REPLY:
Thank you for this observation. We agree that the reference provided is somewhat outdated. Therefore, we have replaced this with a reference to a more recent and very good review of the global incidence of Salmonella illnesses. The amendment in the manuscript is as follows:
Line 37: ...causing an estimated 180 million diarrheal illnesses each year [1].
Thank you also for suggesting CDC data as a source. We have looked into this data, but since it is more focused on illnesses in the USA, we have chosen the following:
Line 458. Besser, J.M. Salmonella epidemiology: A whirlwind of change. Food Microbiol. 2018, 71, 55-59. doi: 10.1016/j.fm.2017.08.018.
Line 71-73 - Is this the first report of this kind of study or are there any studies done previously? The authors have mentioned these studies as lacking, they need to cite any corelating reference if available although the authors have mentioned the presence of Salmonella in H. rugulosus in Discussion (Line 262-276) but I would suggest to cite more reference if published.
REPLY:
Thank you for this question. This is indeed the first report of this kind of data. It is therefore not possible to mention previous studies. The presence of Salmonella in H. rugulosus in the Discussion refers to animals that were collected from frog farms and not from wet markets, therefore they were not frogs specifically destined for human consumption.
We agree that it may be insufficiently explained that this study is essentially a pilot study to collect baseline data. To clarify, the following amendments have been made:
Line 76: Therefore, this pilot study aimed to assess the occurrence of Salmonella in H. rugulosus.
Line 410: Therefore, essentially, this study is a pilot study to collect some baseline data.
Line 94 – How many swabs per frog were collected?
REPLY:
One swab per frog was collected.
Line 113 - Please provide a supplemental file with the genome assembly statistics and metadata for WGS.
We have additionally provided metadata for the WGS in a new supplemental File S1. This is mentioned as follows:
Line 260: Additional data is provided in supplemental File S1.
Line 435: File S1: Metadata of antimicrobial resistance genes collected by whole genome sequecing of 67 Salmonella isolates.
Further details on the genomes of each isolate are available under the accession numbers of the individual isolates, as listed in Table 2.
Line 115 – Was it DNA Blood and Tissue kit or DNeasy kit from Qiagen?
REPLY:
We used the following kit: DNeasy Blood and Tissue Kit (Qiagen, Hombrechtikon, Switzerland)
To clarify, the spelling has been amended:
Line 128: DNeasy Blood & Tissue Kit
Line 117 – Which reagent kit from Illumina did the authors use?
REPLY:
We used the Nextera DNA Flex Sample Preparation Kit (Illumina, San Diego, CA, USA), please see Line 129.
In Line 19, 80-82, authors mention that frogs were collected from Hong Kong markets whereas in Line 160-162 authors mention origination of frogs from mainland China and Thailand. These line lacks clarity, please clarify.
REPLY:
The frogs were collected from the wet markets. The vendors in the wet markets acquire the frogs from distributors who trade frogs that are harvested in farms in mainland China or Thailand.
To clarify, the following amendment has been made:
Line 180: Frogs purchased in wet markets A-H were imported from mainland China...
Table 1: Please mention the wet market names, city, and state as a footnote. Also, what does isolate ID mean, are all these from Farm F?
REPLY:
Thank you for this suggestion. For reasons of data protection, we do not mention the names and precise locations of the wet markets.
The F in the denotation of the isolates refers to "Frog” and is followed by a number. For the wet market that is designated as market F, the F stands alone and is therefore distinct from the isolate ID.
Figure 1: Please indicate the sub clusters in the list given and what sub cluster is the purple group?
REPLY:
The subclusters are indicated within the Figure 1. To clarify that there are eight subclusters, this is now additionally mentioned in the figure legend:
Line 237: subclusters 1-8 are indicated.
The purple group are isolates belonging to S. Hvittingfoss, but these form a single cluster and is not divided in subclusters.
Line 237 – 250 - in silico and n= should be in italics throughout the manuscript
REPLY:
"in silico" has been put in italics throughout the manuscript.
"n=" has been put in italics throughout the manuscript.
Section 3.5 and 3.6 - It would be nice if the authors briefly presented a table of sidewise comparison of phenotype and genotype (detected resistance genes profile) results from all the salmonella isolates. If the authors add the concordance between phenotypic detected resistance and predicted resistance from genomes then, it will make a good addition.
REPLY:
Thank you for these suggestions. During the revision of the manuscript, Table 2 was improved, and now gives a better overview of the genotypes and the corresponding phenotypic resistance profiles of each isolate. As suggested by Reviewer #2, we have added a paragraph highlighting the most important observations of concordance between phenotypic detected resistance and predicted resistance from genomes:
Line 276: There was a correlation between phenotypic antimicrobial resistance and the presence of at least one ARG encoding resistance to aminoglycosides, amoxicillin-clavulanic acid, folate pathway inhibitors, phenicols, macrolides, and tetracycline, but no concordance between phenotypic and genotypic resistance to extended-spectrum beta-lactams and nalidixic acid (Table 2).
The authors have found several ARGs in the salmonella isolates, it would be nice if the authors could add the mobile genetic elements that harbor ARG.
REPLY:
Thank you for this interesting comment. We agree that extending the analysis of the WGS data would be very interesting. We have considered the suggestion to include details on mobile genetic elements but have come to the conclusion that it would be beyond the scope of this manuscript to provide further data at this point. Such data will fit very well to a future manuscript focused on the genetic surroundings of ARGs of Salmonella, including mobile genetic elements and in-depth plasmid analysis.
Reviewer 3 Report
The manuscript by Boss et al. valuable, well written and conducted, with an original contribution to the knowledge of the occurrence and antimicrobial susceptibility profile of Salmonella isolates in an edible frog species in China. I support its further processing after appropriate modifications as outlined below:
L2: I would like to suggest the inclusion of antimicrobial resistance term in the manuscript title
L20: “Salmonella” – please add “spp.”
L80: I would like to suggest to the authors to justify, based on a statistic model, the total number of the collected and processed samples. I wonder, why exactly 103, and not 53 or 203?
L85: “… and weight were documented.” – at the end of the sentence, please insert a reference
L97: please insert an appropriate reference for the 2.2. Salmonella isolation and identification subheading
L135-139: to be more informative for the reader, please present the tested antimicrobials according to their group of which they are a part
L147: please replace “… to at least three antimicrobials from different classes” instead of “… to three or more classes of antimicrobials”
L164: “67 (65%)” – when you express overall prevalence values, please insert in brackets the value of 95% confidence interval
L182: please revise the content of the Table 1, it seems to have different font from the text
L204: in the penultimate column of the Table 2, to be more informative for the reader and to be easier to follow the multidrug resistance, please mention firstly the antimicrobial class, and in brackets the antimicrobials
L238: “we observed” – please avoid the using of personal mode formulations, it is not so characteristic for the scientific style.
L251: I would like to suggest the presentation of the Tabel 3 within a supplementary file
L252: please mention “Legend”
L255: the authors must complete the Discussion chapter with the significance of the intermediate resistance strains (e. g. they can easily become resistant strains). Also, please try to highlight the study limitations (e. g. limited number of processed samples)
L350: the authors must complete the conclusion section with future perspectives in this research area. Also, they can mention that the presented results offer useful insight for public health specialists in the management of human infections
Good luck!
Author Response
Authors' point-by-point replies to
REVIEWER #3:
The manuscript by Boss et al. valuable, well written and conducted, with an original contribution to the knowledge of the occurrence and antimicrobial susceptibility profile of Salmonella isolates in an edible frog species in China. I support its further processing after appropriate modifications as outlined below:
L2: I would like to suggest the inclusion of antimicrobial resistance term in the manuscript title
REPLY:
Thank you for this suggestion. The title has been modified to include the terms antimicrobial resistance profiles and virulence factors:
Line 2: Serotypes, antimicrobial resistance profiles and virulence factors of Salmonella isolates in Chinese edible frogs (Hoplobatrachus rugulosus) collected from wet markets in Hong Kong
L20: “Salmonella” – please add “spp.”
REPLY:
“spp.” has been added and the verb has been adjusted accordingly (Line 21).
L80: I would like to suggest to the authors to justify, based on a statistic model, the total number of the collected and processed samples. I wonder, why exactly 103, and not 53 or 203?
REPLY:
Thank you for this interesting comment. The total number of the collected and processed samples is a result of our sampling method, which was convenience based. For this study, a research assistant visited all wet markets in Hong Kong twice and noted those that sell frogs. Of these markets, nine were selected randomly. This approach is due to the lack of data on the prevalence of Salmonella in frogs, and lack of data on frog import and retail volumes in Hong Kong. Therefore, essentially, this study is a pilot study to collect some baseline data.
We agree that this aspect of the study may have been insufficiently addressed. Therefore, several amendments have been made to the manuscript to clarify:
Line 76: ...this pilot study aimed to...
Line 82: All officially registered wet markets in Hong Kong (a total of 94 markets distributed across Hong Kong, including Hong Kong Island, Kowloon, and the New Territories) [18] were visited twice by a research assistant within a period of three months, and markets that sold edible frogs were noted. Of these wet markets, a total of nine(designated A-I) were selected randomly...
Line 406: ...individual markets, which were randomly selected. Third, the study was challenged by the lack of any previous data on the prevalence of Salmonella in frogs, and lack of data on frog import and retail volumes in Hong Kong. Therefore, essentially, this study is a pilot study to collect some baseline data.
Reference:
Line 517...: Food and Environmental Hygiene Department. List of FEHD Public Markets / Cooked Food Markets. Available online: https://www.fehd.gov.hk/english/pleasant_environment/tidy_market/Markets_CFC_list.html (accessed 25 May 2023).
L85: “… and weight were documented.” – at the end of the sentence, please insert a reference
REPLY:
It is not entirely clear why Reviewer #3 requires a reference for this sentence. This sentence describes the physical examination of the frogs at the laboratory which were carried out by a veterinarian. In our opinion, it is not possible to insert a reference here. To avoid confusion, the sentence has been modified as follows:
Line 90: Physical examinations of the frogs were performed by a certified veterinarian...
L97: please insert an appropriate reference for the 2.2. Salmonella isolation and identification subheading
REPLY:
A reference to the isolation method of Salmonella has been added to the paragraph:
Line 104: Salmonella spp. detection was performed using the ISO 6579-1: 2017 method for isolation and identification of Salmonella [20].
Reference:
Line 525: Mooijman, K.A.; Pielaat, A. Kuijpers, A.F.A. Validation of EN ISO 6579-1-Microbiology of the food chain-Horizontal method for the detection, enumeration and serotyping of Salmonella-Part 1 detection of Salmonella spp.Internat. J. Food Microbiol. 2019, 288, 3-12. 10.1016/j.ijfoodmicro.2018.03.022.
L135-139: to be more informative for the reader, please present the tested antimicrobials according to their group of which they are a part
REPLY:
To provide more information to the reader, the classification of the antimicrobials is indicated, as requested by Reviewer #3 The paragraph has been modified as follows:
Line 152: Antimicrobial agents included beta-lactams including penicillins and cephalosporins (ampicillin (AM), amoxicillin/clavulanic acid (AMC), cefazolin (CZ), cefotaxime (CTX), cefepime (FEP)), quinolones including fluroquinolones (nalidixic acid (NA), ciprofloxacin (CIP)), aminoglycosides (gentamicin (G), kanamycin (K), streptomycin (S)), a sulfonamide (sulfamethoxazole-trimethoprim (SXT)), a phosphonic antibiotic (fosfomycin (FOS)), a macrolide (azithromycin (AZM)), a nitrofuran (nitrofurantoin (F/M)), a tetracycline antibiotic (tetracycline (TE)), and an amphenicol (chloramphenicol (C)) (Becton, Dickinson, Heidelberg, Germany). The MICs of the carbapenems ertapenem....
L147: please replace “… to at least three antimicrobials from different classes” instead of “… to three or more classes of antimicrobials”
REPLY:
The change has been introduced to the manuscript as suggested by Reviewer #3:
Line 166: Multidrug resistance (MDR) was defined as resistance to at least three antimicrobials from different classes[28].
L164: “67 (65%)” – when you express overall prevalence values, please insert in brackets the value of 95% confidence interval
REPLY:
For the overall prevalence values, the 95% confidence interval (CI) values have been inserted, as suggested by Reviewer #3:
Line 21: 67 (65%, CI: 0.554–0.736)
Line 184: 67 (65%, CI: 0.554–0.736)
L182: please revise the content of the Table 1, it seems to have different font from the text
REPLY:
Thank you for this observation. The font in Table 1 has been adjusted.
L204: in the penultimate column of the Table 2, to be more informative for the reader and to be easier to follow the multidrug resistance, please mention firstly the antimicrobial class, and in brackets the antimicrobials
REPLY:
The penultimate column in Table 2 has been amended and indicates the antimicrobial class, and in brackets the antimicrobial.
The footnote has been modified to clarify the resistance profile:
Line 227: b Resistance profiles show the antimicrobial class, and in brackets the individual antimicrobials.
L238: “we observed” – please avoid the using of personal mode formulations, it is not so characteristic for the scientific style.
REPLY:
Thank you for this comment. The terms "we observed" and "we found" have been replaced with the passive voice, as suggested by Reviewer #3:
Line 263: Among the 67 Salmonella isolates, resistance to ampicillin (n=16; 24%), amoxycillin/clavulanic acid (n=6; 9%), cefazolin (n=6; 9%), nalidixic acid (n=24; 36%), ciprofloxacin (n=15; 22%), gentamicin (n=1; 1.5%), kanamycin (n=7; 10%), streptomycin (n=9; 13%), sulfamethoxazole-trimethoprim (n=8; 12%), azithromycin (n=5; 7%), tetracycline (n=38; 57%), and chloramphenicol (n=7;10%) was observed (Table 2)
Line 269: Notably, intermediate resistance to cefotaxime (n=1; 1.5%), cefepime (n=1; 1.5%), nalidixic acid (n=17; 25%), ciprofloxacin (n=52; 78%), streptomycin (n=2; 3%), nitrofurantoin (n=1; 1.5%), and chloramphenicol (n=2; 3%) was found (Table S2).
L251: I would like to suggest the presentation of the Tabel 3 within a supplementary file
REPLY:
Table 3 has been moved to the supplementary file as Table S2, as suggested by Reviewer #3.
L252: please mention “Legend”
REPLY:
This part of the manuscript, consisting of Table 3, has been moved to the supplementary material, as suggested by Reviewer #3, please see above.
L255: the authors must complete the Discussion chapter with the significance of the intermediate resistance strains (e. g. they can easily become resistant strains). Also, please try to highlight the study limitations (e. g. limited number of processed samples)
REPLY:
Thank you for pointing this out. An amendment has been made to highlight the importance of intermediate resistance strains:
Line 385: Likewise, isolates showing intermediate resistance, i.e., reduced susceptibility, are of concern to public health. Reduced susceptibility is important because it facilitates the selection of isolates with higher-level resistance and contributes to the development and spread of resistance to critically important antimicrobial agents such as ciprofloxacin [46,47].
The limitations of this study have been addressed in the last paragraph of the manuscript (lines 396-408). For the revision, some amendments have been made to further highlight the limitations, in particular those related to the sampling strategy:
Line406: ... individual markets, which were randomly selected. Third, the study was challenged by the lack of any previous data on the prevalence of Salmonella in frogs, and lack of data on frog import and retail volumes in Hong Kong. Therefore, essentially, this study is a pilot study to collect some baseline data.
L350: the authors must complete the conclusion section with future perspectives in this research area. Also, they can mention that the presented results offer useful insight for public health specialists in the management of human infections
REPLY:
Thank you for these useful suggestions.
We have amended the discussion and the conclusion sections with the following:
Line 418: To this end, the results presented in study offer useful information for specialists in the public health and the food safety sectors.
Line 422: The role of edible frogs and derived meat products, e.g. frog legs, as sources of antimicrobial resistant and susceptible Salmonella and other foodborne pathogens needs to be further investigated in future.